# Three-Dimensional Volume Rendering in Computed Tomography for Evaluation of the Temporomandibular Joint in Dogs

**DOI:** 10.3390/ani13203231

**Published:** 2023-10-17

**Authors:** Manuel Novales, Rosario Lucena, Eduardo M. Hernández, Pedro J. Ginel, Jesús M. Fernández, Beatriz Blanco

**Affiliations:** 1Department of Animal Medicine and Surgery, University of Córdoba (Spain), Campus de Rabanales, Crtra Nacional IV A, Km 396, 14014 Córdoba, Spain; pv2lusor@uco.es (R.L.); pv1gipep@uco.es (P.J.G.); pv9blnab@uco.es (B.B.); 2Department of Animal Medicine and Surgery, University Complutense of Madrid (Spain), Avda. Puerta de Hierro s/n, 28040 Madrid, Spain; jesusmfe@ucm.es

**Keywords:** canine TMJ, CT scan, multidetector computed tomography, volume rendering, three-dimensional image

## Abstract

**Simple Summary:**

The temporomandibular joint (TMJ) of dogs is a bilateral joint mainly used for chewing, and subsequently, an area where pathologies are frequently seen. Computed tomography (CT) represents a very important imaging modality used to diagnose TMJ diseases in dogs. A three-dimensional representation of CT images created by a volume rendering method (3DVR) could improve our understanding of the alterations in this joint. In this study, we assessed 3DVR as an ancillary method for the description and diagnosis of TMJ pathologies in dogs.

**Abstract:**

Based on computed tomography (CT) images, volume rendering was used to obtain a three-dimensional representation of data (3DVR). The aims of this study included: describing the bone anatomy of the temporomandibular joint (TMJ) of dogs; comparing the TMJs of each dog by skull type and age; comparing 3DVR images with three-standard-plane CTs; assessing soft tissues adjacent to the TMJ and assessing pathological cases. Multidetector computed tomography scans of bilateral TMJs of 410 dogs were observed. From a ventral view, slight displacements in the positions of the skulls were seen, whereas from a caudal view, differences in amplitude of the articular space were observed. Dolichocephalic and mesaticephalic dogs showed more similar TMJ features than brachycephalic dogs. The shape of the TMJ bones were irregular in dogs under 1 year old. The 3DVR images related to the three-standard-plane CT improved the overall comprehension of the changes in the articular space amplitude and condylar process morphology. The fovea pterygoidea, mandibular fossa and retroarticular process were perfectly shown. A better spatial situation of adjacent soft tissues was obtained. The 3DVR represents an ancillary method to the standard-plane CT that could help in the understanding of the anatomy and diagnoses of different pathologies of the TMJ in dogs.

## 1. Introduction

The temporomandibular joint (TMJ) of dogs is a bilateral synovial condylar joint that plays an important role in chewing, swallowing, oral health and nutrition [1]. This joint is composed of two main structures: the condylar process of the mandible and the mandibular fossa of the temporal bone. The condylar process is a transverse and sagittal elongated convex articular process that articulates with the mandibular fossa of the squamous temporal bone [2]. A thin cartilaginous disc lies between the cartilage-covered articular surface of the condylar process of the mandible and the cartilage-covered mandibular fossa of the temporal bone [2]. This disc divides the joint into two cavities: dorsal (or temporal) and ventral (or mandible) [3]. The cartilaginous disc is fibrous and thin with an evident rostral thickness to avoid anterior luxation during substantial vertical movements [4]. This joint is covered by a capsule attached around the joint surfaces, with synovial fluid inside. The capsule is reinforced by the temporomandibular lateral ligament and stabilized by powerful masticatory muscles [2]. This joint is subject to frequent pathologies in dogs due to its motility and complex morphology. Most of these pathologies can be diagnosed using scan CT; thus, it is an excellent method to study TMJ bone elements from different perspectives [5,6].

Computed tomography is a cross-sectional imaging modality based on the absorption of patient X-rays [7]. This imaging technique combines X-rays and a software to develop cross-sectional views of the body without creating any overlapping anatomic structures [8]. Recently, the introduction of multi-detector computed tomography (MDCT) has greatly improved the volumetric reconstruction using real isotropic voxels [7,8,9,10].

The transverse (axial), sagittal and dorsal planes are considered the *standard* planes for displaying [11]. Through a process called *multiplanar reformation (MPR)*, data are stacked along this axis to create a volume, and then re-sliced along the other planes to create sagittal, dorsal or oblique scans [7,12,13].

Acquired CT data can be used to create a *volume rendering (VR) image*, which is a three-dimensional representation of data [13,14,15,16,17,18]. VR assigns opacity values on a full spectrum from 0 to 100% (total transparency to total opacity) using a variety of computational techniques [12]. VR achieves a similar 3D appearance to allow visualization of the bone surfaces from a relatively natural anatomic perspective [12]. Although for most authors, the initials VR stand for 3D images [13,15,19], 3DVR is considered the correct terminology for use in human [15,18,20,21] and veterinary medicine [6,19].

Images obtained using 3DVR can be reformatted by a computer program called *segmentation*, which is the process of selecting data to be included in a 3D image. *Region- of- interest editing* is the most basic method of segmentation. A region of interest is manually removed, drawing a rectangular, elliptical or other shape using a sort of virtual scalpel to “cut” the defined region [12,21]. This method allows for the visualization of anatomical details that would be difficult to evaluate using transverse (axial) reconstructions alone [11]. Automated segmentation programs are also available involving placement of a “seed”, then expanding the region to be included or excluded using threshold-based algorithms [12].

In veterinary scientific literature, descriptions of normal dog TMJs using a CT scan can be found in a PhD project [22], as well as in books containing detailed descriptions of pathologies of TMJ showing transverse images or MPR using CT scans [5,6]. However, 3DVR images of TMJs are only briefly included when studying CT scans of different TMJ pathologies [6,23,24,25]. Moreover, in specific works about TMJ, only lateral, rostrolateral [6,25,26,27] and ventral images [27] of TMJ were superficially shown. No detailed studies of the medial, ventral, ventrolateral and caudoventral aspects of the TMJ and its relationship with other bone structures were found to employ 3DVR images.

The objective of this study was to show that 3DVR images could provide excellent anatomic detail of the bone structures of TMJ in dogs, supporting the interpretation of CT standard planes (transverse, sagittal and dorsal) and better understanding of joint anatomy. To obtain this objective, we focused on four points: (1) Providing a detailed description of the bone anatomy of TMJ using 3DVR images; (2) Assessing each TMJ according to the skull type and age of dogs using 3DVR images; (3) Assessing each TMJ by comparing 3DVR images with the 3-standard-plane CT; (4) Providing a pictorial essay of 3DVR images in some pathological cases.

## 2. Materials and Methods

The Diagnose Imaging Service database at the Clinical Veterinary Hospital of the University of Córdoba (Spain) was searched for CT scans of the skulls of dogs from January 2021 to June 2023. A total of 410 CT scans were studied: 233 male (56.8%) and 177 female (43.2%) dogs, aged between 2 and 12 years. According to previously described skull types [2], dogs were grouped into dolichocephalic (*n* = 6; 1.46%), mesaticephalic (*n* = 348; 84.87%) and brachycephalic (*n* = 56; 13.65%) groups.

Dogs underwent skull CTs for a variety of reasons, including: nasal cavity and frontal sinus pathologies (*n* = 88; 21.5%); orbital and salivary glands (*n* = 13; 3.17%); external, middle and inner ears (*n*= 38; 93%); calvarium and encephalic (*n* = 187; 45.6%) diseases; and oral cavity and masticatory apparatuses, including temporomandibular joints (*n* = 84; 20.5%). Only 10 animals were studied for specific problems in the temporomandibular joint.

All dogs were scanned using multidetector computed tomography (32 slice helical scanner) (Revolution ACT; General Electric Health Care, Beijing, China). The standard protocol was 120 kVp and 100–300 mA, with a 512 × 512 acquisition matrix. Sequential transverse CT slices of 0.6 (spiral pitch factor of 1) and 1.25 mm thicknesses (spiral pitch factor of 0.75) were obtained [22,27].

Dogs were placed in a sternal decubitus position with the mandibles parallel to the table using positional aids. Dogs were restrained using general anesthesia, premedicated with methadone (0.3 mg/kg IM) and dexmedetomidine (0.003 mg/kg, IM); anesthesia was induced using dose-effect propofol (IV) and maintained with isofluorane at 100%. Dogs’ mouths stayed slightly opened to place the endotracheal tube.

A contrast CT using iohexol (2 mL/kg; Omnitrast 300, Lab. Schering, Berlin, Germany) injected in the cephalic vein was performed after a plain CT to evaluate soft tissues.

All images were transferred to a DICOM workstation, and then CT images were analyzed with an open-source media image viewer Horos© (open-source code software (FOSS, v.3.3.6) program distributed free of charge under the LGPL license at Horosproject.org and sponsored by Nimble CO LLC d/b/a Purview in Annapolis, MD, USA).

An evaluation of the whole skull of each dog was performed using the 3 standard planes of CT (transverse, sagittal and dorsal). Images were assessed following algorithms of bone (window width, WW > 1500 UH; window level, WL: 400–500 UH) and soft tissues (WW: 400–500 UH; WL 40–50 UH) [28]. Resulting 3DVR images were generated to obtain a complete view of the dog’s TMJ. Each 3DVR image could be rotated in 360°.

Images obtained using 3DVR were reformatted by segmentation, as shown in Figure 1. The lateral view of the image, obtained with the left and right mandible completely overlapping, was first transversely segmented rostral and caudal to the ramus of the mandible (Section A). The obtained images were rotated to obtain a rostral view of the skull, and then a sagittal segmentation was performed (Sections B and C) to show a detailed image of both TMJs (Sections D and E). In these views, the medial aspect of the joint was shown.

All images were independently examined by two radiologists with more than 15 years of experience in performing and reading CT scans. Results were revised for consensus by participation of a third radiologist. The evaluation was carried out following established criteria for normal (22) and abnormal TMJs (5,6).

## 3. Results

### 3.1. A Detailed Description of the Bone Anatomy of TMJ Using 3DVR Images

The 3DVR images were segmented rostral to the ramus of the mandible to highlight the different anatomic structures of each aspect of the joint.

From the *lateral* view, it is possible to see the relationship between the TMJ and ramus of the mandible. The ramus of the mandible corresponds to the caudal non-tooth-bearing vertical part of the bone. It contains three processes: the coronoid process, which forms the most dorsal part of the mandible; the condylar process, which is transversely elongated, forming the temporomandibular joint by its articulation with the mandibular fossa of the squamous temporal bone; and the angular process, which, in a caudoventral position, serves as an attachment for the masseter and digastricus muscles [2]. The image showed the proximity of the TMJ to the tympanic bullae and hyoid apparatus. The *ventrolateral* view provides a good, detailed image of the mandibular fossa and its caudoventral extension, the retroarticular process (Figure 2).

The *rostral* view of the TMJ clearly shows the rostromedial-caudolateral orientation of the ramus of the mandible, dividing the joint into two approximately equal parts. The condylar process has a dorsolateral-ventromedial orientation. The medial and lateral borders have different morphologies. The *rostromedial* view clearly shows the dorsal articular surface of the condylar process and the corresponding surface of the mandibular fossa. The condylar process is slightly concave in the rostral margin of the dorsal articular surface. The joint space is wider in the rostral portion of the joint (Figure 3).

The *rostrolateral* view of the TMJ mainly shows the masseteric fossa and zygomatic arch. From the *caudal* view, the ventral part of the base of the zygomatic process expands to form a transversely elongated, smooth area, called the mandibular fossa, which receives the condyle of the mandible to form the TMJ. The retroarticular process is a ventral extension of the squamous temporal bone (Figure 4).

The *ventral* view shows the relation of the TMJ with the wall of the tympanic bulla being caudoventrally surrounded by the stylohyoid bone. In this position, the angular process divides the mandibular fossa into two sections: lateral and medial. The medial side highlights the retroarticular process in the caudoventral aspect of the joint. The *caudoventral* view rostrally displaces the angular process to offer a better view of that joint, thus exposing a clearer view of the neck of the condylar process (Figure 5).

### 3.2. Comparison by 3DVR of TMJs in Each Dog, According to Skull Type and Age

The joint assessment of both TMJs by 3DVR were used to detect asymmetry in the positioning of the skull, differences in joint amplitude and relations with adjacent bone structures, according to the type of skull (dolichocephalic, mesaticephalic or brachycephalic) [2] and age of the dog.

The 3DVR images were segmented rostral to the ramus of the mandibles to compare rostral and caudal aspects of both TMJs. From the rostral view, the cuff of the endotracheal tube is observed. From this view, the pterygoid hamulus, bone of the hyoid apparatus and tympanic bullae are visualized. In a normal dog, the stylohyoid bone is curved to adjust to the ventromedial wall of the tympanic bulla. From a caudal view, it is possible to evaluate the asymmetry in skull position and even the differences in joints that may be produced under general anesthesia (Figure 6).

Regardless of breed, skulls of dolichocephalic (Figure 7) and mesaticephalic dogs (Figure 8) had more similar TMJ features than skulls of brachycephalic dogs (Figure 9), which showed more marked differences between TMJs. Brachycephalic dogs showed flatter TMJs, especially in the medial aspect, and the retroarticular processes did not encircle the condylar processes.

Related to age, the articular spaces in TMJ were wider in younger dogs under 6 months of age than in adult ones; bone margins seemed irregular in shape in dogs under 12 months old.

### 3.3. A Detailed Assessment of Each TMJ Comparing 3DVR Images with the Three Standard CT Planes

The 3DVR images were segmented rostral to the ramus of the mandibles and the obtained image was rotated to obtain a rostral view of the skull. Then, a sagittal segmentation was performed. From this view, the right aspect of each TMJ could be individually assessed. The *lateral* view of each TMJ revealed that the most rostral section of the temporal bone was thinner, and the articular space was wider. The *medial and rostromedial* views showed the location of the fovea pterygoidea, the area of termination of the pterygoideus lateralis muscle. This muscle is inserted in the medial surface of the condyle of the mandible, just ventral to its articular surface. This detail explains how transverse planes with a difference of just 0.5 mm or less may offer a different morphology of the TMJ in transverse planes (Figure 10).

In mesaticephalic healthy dogs, changes in transverse images were evident when the TMJ was assessed in transverse planes, with differences of 0.5 cm following a rostrocaudal direction. These included: (1) Alteration of the thickness of the subchondral temporal bone (thicker in the more caudal aspect); (2) Changes in the size of the articular space (narrower in the more caudal aspect); (3) Changes in the shape of the medial aspect of the condylar process (turning to a flat shape in the more rostral aspect). These changes can be seen in adult (Figure 11) and young dogs (Figure 12).

### 3.4. A Pictorial Essay of 3DVR Images in Some Pathological Cases

Four pathological cases were selected to assess the TMJ by 3DVR, including an abnormal joint adjustment (Figure 13), subchondral bone alterations (Figure 14), cysts and osteoarthritis of the TMJ (Figure 15) and cysts in the nasopharyngeal passage, which changed the position of the TMJ, inducing asymmetry of the articular space (Figure 16).

## 4. Discussion

In this study, a detailed description of the bone anatomy of the TMJs in dogs is provided using 3DVR images. Our aim was to show that 3DVR images represent an ancillary method to the standard planes in CT to ease understanding of the anatomy and diagnoses of the different pathologies of the TMJ in dogs. Using 3DVR provides excellent images of all the bone components of the joint, focusing on the shape and spatial relationships among the different bone structures [1,17,18]. It also helps to determine the extension of anomalies to plan for the best surgical management [29]. However, as some artifacts have also been observed when studying 3DVR images [7], it remains mandatory to compare 3DVR images with standard plane images.

Several factors must be considered to obtain high-quality and high-resolution images using 3DVR. In our study, we used 120 kV and 100–300 mAs, with slices of 0.6 and 1.25 mm and a pitch of 1 and 0.75, respectively. In another study of dogs using CT images of the TMJ [23], the authors employed 100 kV and 100 mAs, with a slice thickness of 1.1 mm and pitch of 0.75, as their technical parameters. However, in this work, no 3DVR images of the TMJ were shown for comparative purposes. In a human study of 3DVR images of the mandible, it was shown that slices of 0.6 mm were usually used for the diagnosis of mandible pathologies, but slices up to 1.25 mm also resulted in images of good quality [20]. In our own experience, slices of 0.6 mm with a pitch of 1 and slices of 1.25 mm with pitch of 0.75 were both appropriate to obtain high-quality 3DVR images. The image quality also depends on the pitch [30]. In our study, a pitch of 0.75 was selected to overlap planes and improve the image resolution for reconstruction. No studies using different pitches were found to compare our results.

In the present study 3DVR images of the TMJ were greatly descriptive, showing every detail of the TMJ. Until now, 3DVR images of TMJ have only been briefly studied when focusing on different alterations in craniomandibular osteopathy, dysplasia [6], mandible luxation [6], post-traumatic ankylosis [23], sarcoma in the caudal portion of mandible [24], mandible osteomyelitis [25], fractures secondary to septic arthritis or osteomyelitis [25] or even to fit the site of puncture of a joint [31]. These studies showed lateral and rostrolateral views of the TMJ, but only Mielke et al. 2017 [29] showed ventral images of the TMJ of dogs of different breeds, relating its anatomical positions with tympanic bullae. However, no detailed studies employing 3DVR images to describe the medial, ventral, ventrolateral and caudoventral aspects of the TMJ and its relationship with other bone structures have been found in the literature.

Through the joint assessment of both TMJs by 3DVR, slight displacements in the position of the skull were seen, especially in the ventral view, as well as differences in the amplitude of the articular spaces in the caudal view. These changes could be due to anesthesia [32] or from different pathologies. Overall, 3DVR images were useful for the better understanding of these changes that, in some conditions, were more difficult to detect in the standard planes alone. The 3DVR images showed a whole view of the TMJ, where both rostral and caudal sides could be observed in a single image.

Although textbooks on the systematic anatomy of dogs [2] and the bone anatomy of the skulls of dogs [33] usually show images of adult mesaticephalic dogs, differences were found in the TMJ related to the type of skull and breed [8,22], and there were also individual variations [22]. In a large, previous study, dolichocephalic and mesaticephalic type skulls were shown to have very similar TMJs and most of the differences were noted in small brachycephalic breeds [22]. Similarly, in our study, skulls of dolichocephalic and mesaticephalic dogs showed more similar TMJ features than those of brachycephalic dogs. This group showed flatter TMJs, especially in the medial aspect, and had retroarticular processes that did not encircle the condylar processes (Figure 9).

Related to the age of dogs, the TMJ was not fully developed until dogs were 1 year old, and articular spaces were wider in young dogs than in adult ones [5,34]. In our study, dogs under 1 year old had bones of the TMJ that were not fully ossified, and an irregular shape could be observed in the 3DVR images (Figure 12). The study of the influence of the type of skull and age of dogs on 3DVR images of the TMJ were not under the scope of our study. In future studies, these factors could be investigated further.

In our study, a detailed assessment of each TMJ comparing 3DVR images with the three standard CT planes clearly improved comprehension of the changes in the amplitude of the articular space and the morphology of the condylar process, according to the level of the slice. Transverse plane CTs of different pathologies of the TMJ were found [5,6], with occasional dorsal and sagittal planes, although the reformatted dorsal and sagittal planes were of poorer quality than transverse ones [5]. According to our results, it would be interesting to study both 3DVR images as well as the ones in the three standard CT planes to correctly assess the TMJ of dogs. Until now, no studies have investigated this topic.

In our study, the fovea pterygoidea in the medial aspect of the TMJ could be perfectly observed using 3DVR images. The correct observation of this fovea is fundamental to the diagnosis of a specific pathology of the muscle pterygoid lateralis. A bilateral pterygoid myositis ossificans-like lesion in dogs affecting the m. pterygoideus lateralis has been described by CT and by cone beam computed tomography (CBCT) [35], but no detailed images have been reported using 3DVR. Thus, 3DVR images could help in the diagnosis of pathologies of that fovea.

Our detailed study detected changes in the morphology of the condylar process, depending on the level of the slices in the transverse and dorsal planes. In the dorsal and transverse planes of the TMJ, the condylar process appeared as an elongated structure with an irregular shape, being thicker around the midpoint and projecting rostrally and ventrally at this level to join the mandibular ramus. This is similar to previous descriptions but have not been supported by images [8]. In our study, 3DVR images of the medial and rostromedial views clearly showed this process.

In our study, the mandibular fossa and retroarticular process could be observed by 3DVR images, mainly from their lateral, caudal and ventral sides. According to the dog breed, different anatomical conformations of these structures were described in both a schematic representation and in the sagittal planes [8]. As in the standard planes the morphology of the mandibular fossa may change depending on slice level; 3DVR images could help to address these variations (Figure 2, Figure 4, Figure 5 and Figure 11).

The soft tissues adjacent to the TMJ were studied using transverse planes and 3DVR images. Intrinsic soft tissues of the TMJ (articular cartilage, disc, joint capsule and lateral ligament) could not be clearly delineated using CT scans [5,6,36], MDCT or CBCT [8]. MRI scans have been primarily used for the evaluation of soft tissue components of the maxillofacial region, specifically of the TMJ articular disc [6,8,37]. The articular disc could be observed by MRI in 70% of cases [38]. Thus, 3DVR images could be useful in obtaining a better spatial situation of the soft tissues of the TMJ.

When selecting clinical cases, the use of 3DVR helped in the understanding of the TMJ. The abnormal adjustment between TMJs could be better detected using rostral views of the 3DVR images segmented caudally to the ramus of the mandibles than using transverse planes. There was a marked relation between alterations in the subchondral bone in the transverse and dorsal planes and in the 3DVR images. There was also a strong relation between alterations of the subchondral bone of the two bones of the joint. In our experience, the size of the cysts observed in dorsal and transverse planes correlated well with the 3DVR images. The size of the cysts could be measured similarly from the standard planes (transverse and dorsal) and from the 3DVR images.

## 5. Conclusions

Images by 3DVR should be used to assess the TMJ of dogs to improve the interpretation of the standard MPR and help further our understanding of the bone anatomy of this joint.

## Figures and Tables

**Figure 1 animals-13-03231-f001:**
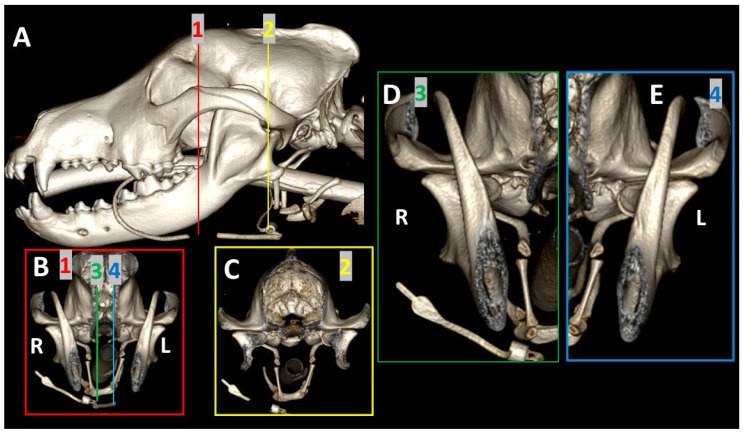
Segmentation of a 3DVR image of the skull of a 6-year-old, female Golden retriever. Section (**A**): lateral view of the skull, with transverse segmentation obtained rostral (red line) and caudal (yellow line) to the ramus of the mandible. Section (**B**): rostral view of the skull obtained at the level of the red line. Section (**C**): rostral view of the skull obtained at the level of the yellow line. Sections (**D**,**E**) represent a magnified image for detail of both TMJs obtained at the level of green (3) and blue (4) lines of Section (**B**).

**Figure 2 animals-13-03231-f002:**
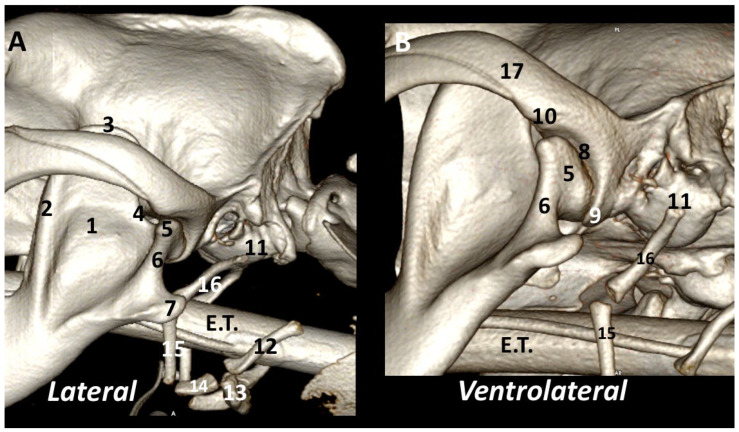
Left TMJ 3DVR images of a 2-year-old, male Labrador retriever from lateral (**A**) and ventrolateral (**B**) views: (1) ramus of the mandible (masseteric fossa); (2) coronoid crest; (3) coronoid process; (4) mandibular notch; (5) condylar process of mandible; (6) neck of the condylar process; (7) angular process of the mandible; (8) mandibular fossa; (9) retroarticular process; (10) protrusion of rostrolateral aspect of mandibular fossa; (11) tympanic bulla; (12) thyrohyoid; (13) basihyoid; (14) ceratohyoid; (15) epihyoid; (16) stylohyoid; (17) zygomatic arch. E.T.: endotracheal tube.

**Figure 3 animals-13-03231-f003:**
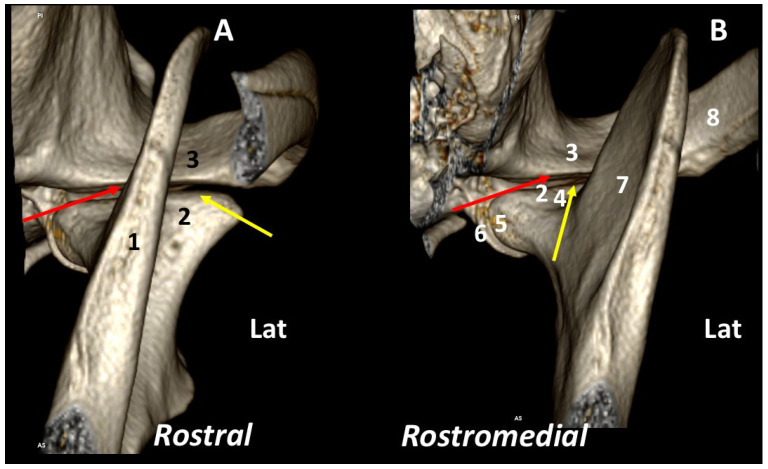
Left TMJ 3DVR images of a 2-year-old, male Labrador retriever from rostral (**A**) and rostromedial views (**B**): (1) ramus of the mandible; (2) condylar process of the mandible; (3) squamous temporal bone; (4) neck of the condylar process; (5) fovea pterygoidea; (6) retroarticular process; (7) ramus of the mandible (medial aspect); (8) zygomatic arch. Surface of the mandibular fossa (red arrows); dorsal articular surface of the condylar process (yellow arrows). (Lat: Lateral aspect).

**Figure 4 animals-13-03231-f004:**
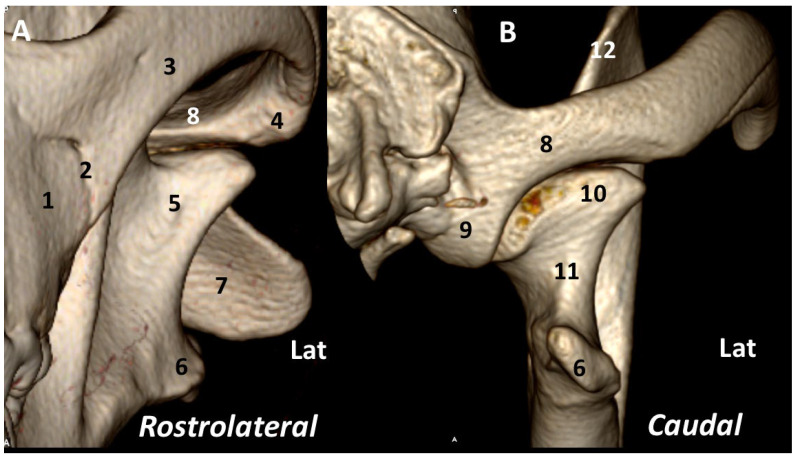
Left TMJ 3DVR images of a 2-year-old, male Labrador retriever from rostrolateral (**A**) and caudal (**B**) views: (1) maxilla; (2) sutura zygomaticomaxillaris; (3) zygomatic arch; (4) protrusion at rostrolateral aspect of mandibular fossa; (5) masseteric fossa; (6) angular process of the mandible; (7) axis; (8) squamous temporal bone; (9) retroarticular process; (10) condylar process; (11) neck of the condylar process; (12) coronoid process. (Lat: lateral aspect).

**Figure 5 animals-13-03231-f005:**
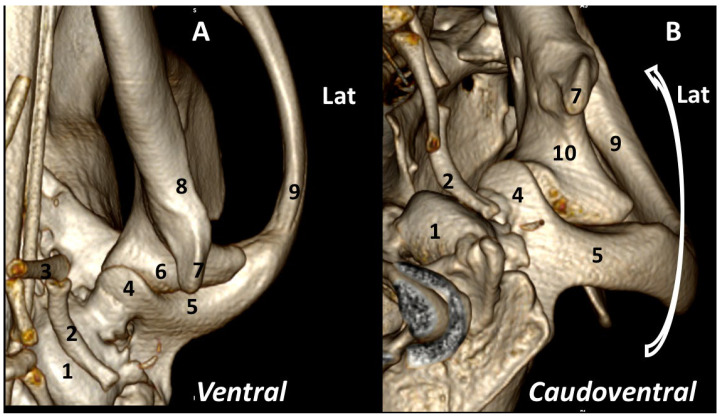
Left TMJ 3DVR images of a 2-year-old, male Labrador retriever from ventral (**A**) and caudoventral (**B**) views: (1) tympanic bulla; (2) stylohyoid; (3) epihyoid; (4) retroarticular process; (5) squamous temporal bone; (6) condylar process; (7) angular process; (8) ramus of mandible; (9) zygomatic arch; (10) neck of condylar process of the mandible. (Lat: Lateral aspect).

**Figure 6 animals-13-03231-f006:**
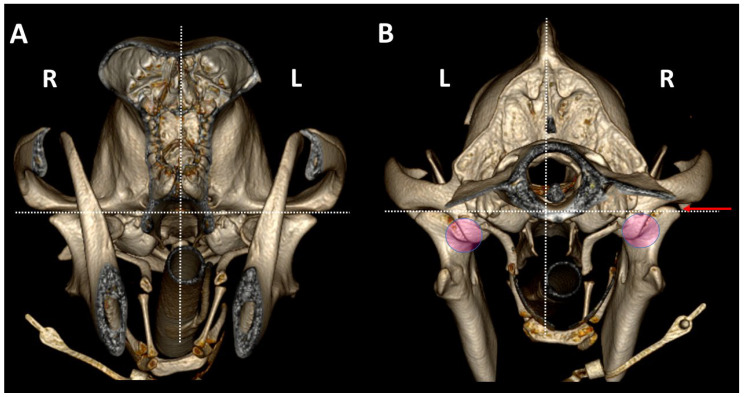
TMJ 3DVR images of a 6-year-old, female Golden retriever, showing the *influence of skull symmetry between TMJs, from* rostral (**A**) and caudal (**B**) views. Although rostrally, there appears to be good symmetry between joints, from a caudal view, it is possible to see that the caudal part of the skull is displaced to the right side (R). The right TMJ shows a joint space 0.5 mm wider than the left TMJ (red arrow). In this animal, the retroarticular process caudomedially encircled the medial aspect of the condylar process (circles). (L: left side; R: right side). Dotted line show: the dorsal planes (horizontal planes) at the level of both TMJs and the midsagittal plane (vertical lines).

**Figure 7 animals-13-03231-f007:**
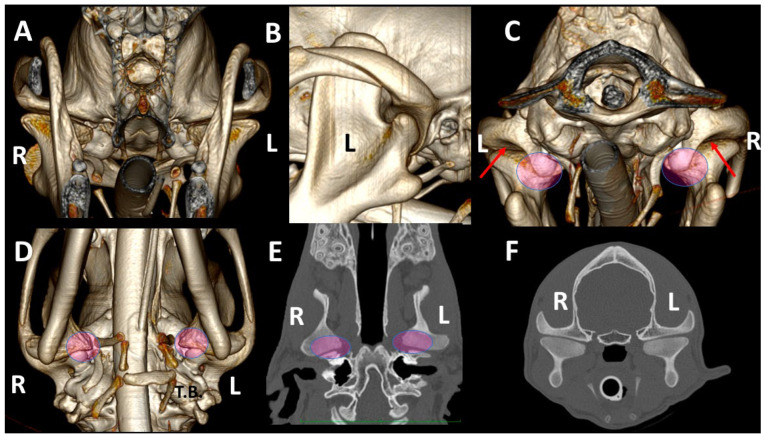
TMJ 3DVR images of a 10-month-old, male Greyhound with a *dolichocephalic skull, from* rostral (**A**), left lateral (**B**), caudal (**C**) and ventral (**D**) views, as well as dorsal (**E**) and transverse (**F**) planes. The images show straight zygomatic arches and tympanic bullae near the TMJs. We see wide joint spaces (red arrows) and condylar processes have scarce osseus densities without clear definition of its subchondral bones. (L: left side; R: right side). Pink circles show the medial part of the TMJ in different views.

**Figure 8 animals-13-03231-f008:**
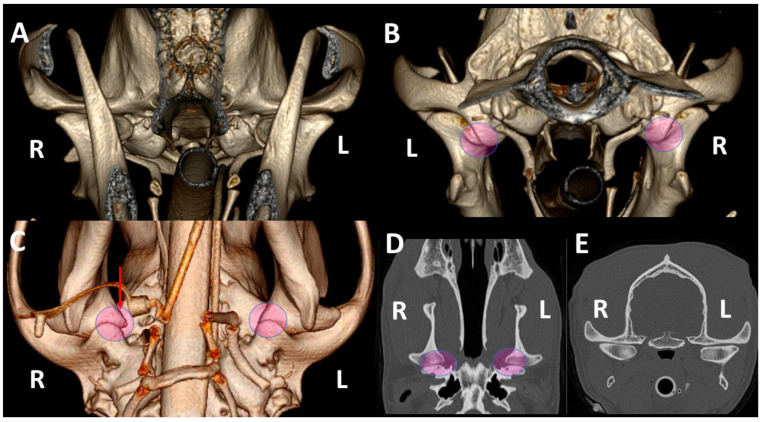
TMJ 3DVR images of a 2-year-old, female Golden retriever with a mesaticephalic *skull*, from rostral (**A**), caudal (**B**) and ventral (**C**) views, as well as dorsal (**D**) and transverse (**E**) planes. Both retroarticular processes caudomedially encircled the condylar processes (circles). The right condylar process (red arrow) is slightly displaced medially; this displacement is difficult to perceive in dorsal and transverse planes. The dog has normal osseous density and the subchondral bones are well-ossified. (L: left side; R: right side).

**Figure 9 animals-13-03231-f009:**
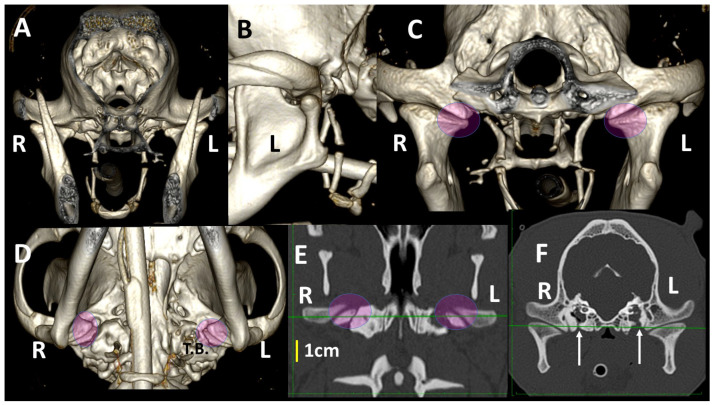
TMJ 3DVR images of a 1-year-old, female French bulldog with a *brachycephalic skull*, from rostral (**A**), left lateral (**B**), caudal (**C**) and ventral (**D**) views, as well as dorsal (**E**) and transverse (**F**) planes. These images show flatter TMJs, especially in the medial aspects. The retroarticular processes do not encircle the condylar processes (circles). Both tympanic bullae show fluid density, secondary to chronic otitis (L: left side; R: right side). Green lines show both TMJ at the same levels in (**E**,**F**). White arrows show a previous surgery in both tympanic bulla.

**Figure 10 animals-13-03231-f010:**
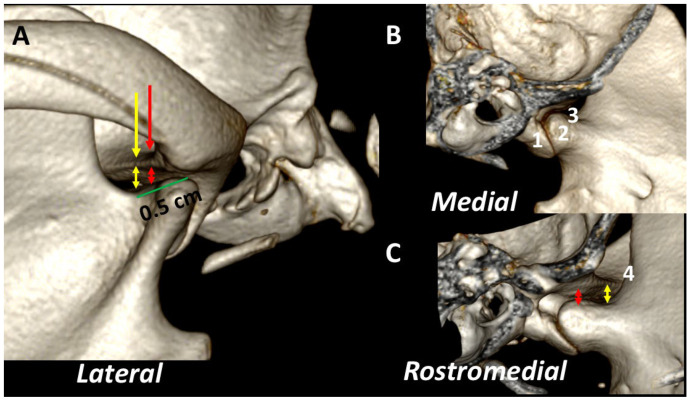
Detailed 3DVR image of the left TMJ of a 2-year-old, male Labrador retriever from lateral (**A**), medial (**B**) and rostromedial (**C**) views. The temporal bone is thinner in the more rostral aspect (yellow arrow) than in the caudal one (red arrow). The joint space is wider rostrally (red and yellow double-sided arrows). (1) Retroarticular process; (2) fovea pterygoidea; (3) neck of the condylar process; (4) mandibular notch.

**Figure 11 animals-13-03231-f011:**
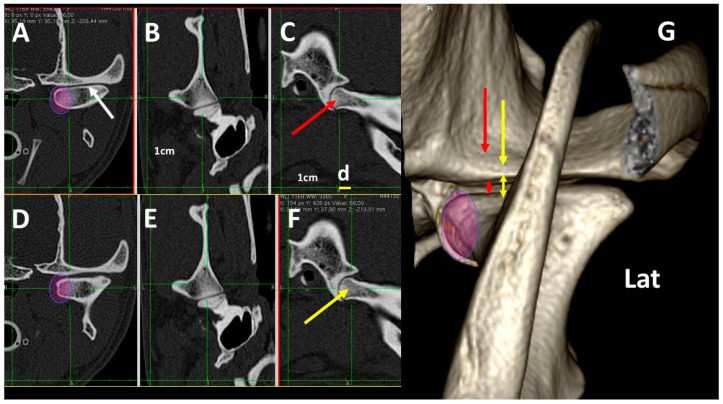
Normal left TMJ 3DVR image of a 2-year-old, male Labrador retriever from a rostral view, showing *morphologic differences in the transverse planes, according to *transverse (**A**,**D**), dorsal (**B**,**E**) and sagittal (**C**,**F**) planes. In each horizontal row, green lines show the level of each transverse plane. (**D**) is obtained 0.5 mm (d) more rostrally than (**A**) (d). Depending on the level, the morphology of the condylar process changes in its medial aspect (circle), and there is a change in the joint space. The 3DVR image (**G**) shows the morphology of the medial aspect of the condylar process (circle) and how the joint space is wider in rostral portions of the joint (red and yellow arrows). The white arrow shows subchondral bones with normal density. (Lat: lateral aspect). It is possible to distinguish the different amplitude of the TMJ in caudal (red double skulled arrow) and cranial (yellow double skulled arrows) part of the joint.

**Figure 12 animals-13-03231-f012:**
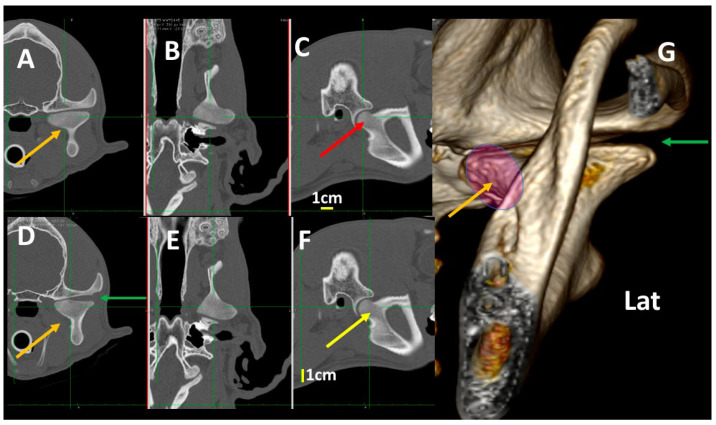
Normal left TMJ 3DVR images of a 4-month-old, male crossbreed with a mesaticephalic skull showing morphologic differences in the transverse planes, according to transverse (**A**,**D**), dorsal (**B**,**E**) and sagittal (**C**,**F**) planes. Left TMJ 3DVR image in rostral view (**G**). In each horizontal row, green lines show the level of each transverse plane. (**D**) is obtained 0.3 mm (d) more rostrally than (**A**) (d). The condylar process is not fully ossified and has an irregular aspect (brown arrows). The joint space is wider than in an adult dog (green arrow). In the condylar process, it is not possible to distinguish the density of the subchondral bones from the rest of the bone (Lat: lateral aspect). In each horizontal row, green lines show the level of each transverse plane. The slice in (**C**) (red arrow) is obtained 0.3cm more caudally than slice in (**F**) (yellow arrow). Pink circles shows the medial aspect of the condylar process.

**Figure 13 animals-13-03231-f013:**
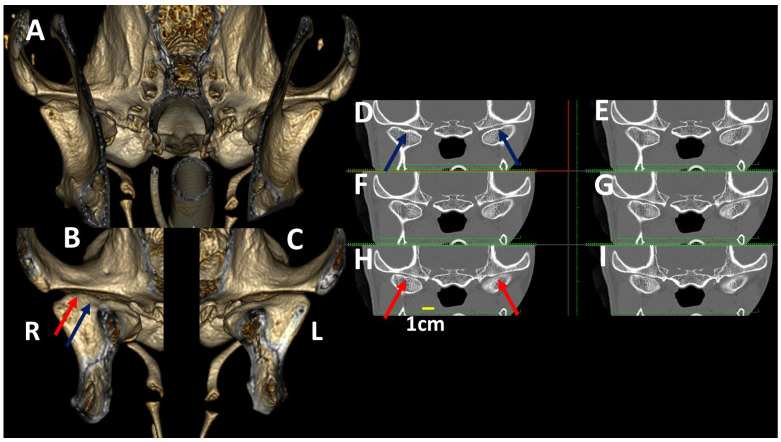
TMJ 3DVR image of a 7-year-old, female crossbreed dog, rostral (**A**) and caudal (**B**) to the ramus of the mandible in the right (**B**) and left (**C**) TMJ, and consecutive transverse planes of the TMJ (**D**–**I**). In the rostral aspect of the joint (blue arrows), the space is wider than in the caudal aspect (red arrows). In the caudal aspect, the space between the temporal bone and lateral aspect of the condylar process is thinner. These details are better seen in (**B**,**C**) than in the transverse planes. This adjustment of the articular surfaces is not usually seen (R: right side, L: left side).

**Figure 14 animals-13-03231-f014:**
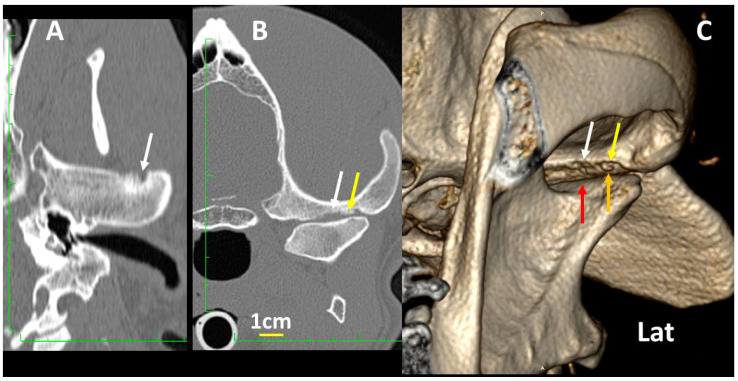
Details of the left TMJ of a 7-year-old, male Mastiff with osteoarthritis, showing dorsal (**A**), transverse (**B**) and 3DVR images of the left TMJ in the rostrolateral view (**C**). The mandibular fossa of the temporal bone shows eroded areas in the subchondral bone (white arrows), and even a small osteochondral fragment (yellow arrows), which correlates with osteoarthritis of the TMJ. In (**C**), it is evident that the most rostral portion of the condylar process (red arrow) is wider than the caudal portion (orange arrow).

**Figure 15 animals-13-03231-f015:**
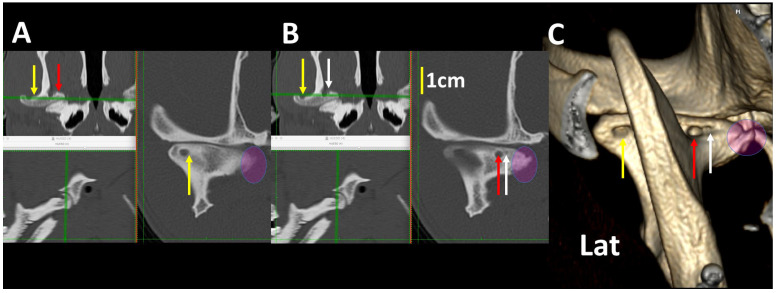
MPR images of the right TMJ at different levels (**A**,**B**) and the 3DVR image of the same joint (**C**) of an 8-year-old, male American Staffordshire terrier. Three cyst-like lesions are observed (correspondent arrows in different colors). The smaller cyst is 0.2 mm in diameter. There was good correspondence between the measurement of the cyst in transversal and 3DVR images. A large osteophyte in the medial aspect of the condylar apophysis is shown (circle), which correlates with osteoarthritis (Lat: lateral aspect).

**Figure 16 animals-13-03231-f016:**
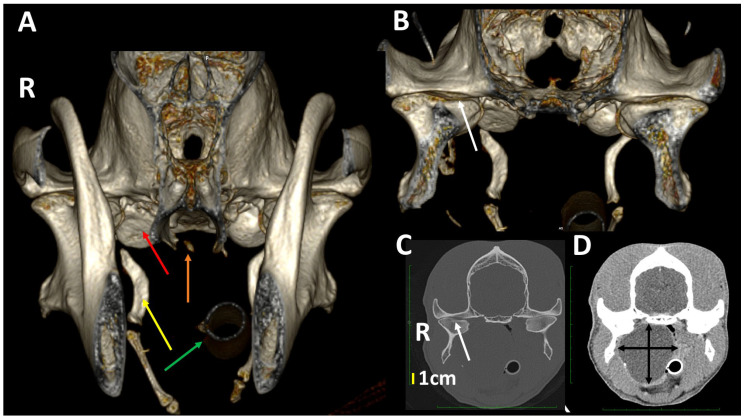
A 3DVR image of the TMJs rostral (**A**) and caudal (**B)** to the ramus of the mandible of a 5-year-old, male German shepherd, as well as transverse planes at the level of the TMJ with bone (**C**) and soft tissue algorithms (**D**). On the right side (R), there is a large chronic nasopharyngeal cyst with several years of evolution. The cyst has produced a lateral displacement of the right TMJ with a slight widening of the joint space (white arrow). There is a flattening of the tympanic bulla (red arrow), a widening of the stylohyoid bone (yellow arrow), a fracture of the pterygoid hamulus (orange arrow) and a right displacement of the nasotracheal tube (green arrow). The lateral displacement of the right TMJ is clearly shown in (**B**,**C**) (white arrow), but not in (**A**). (R: right side, L: left side).

## Data Availability

In this morphological study all important data are included in text and there is no need of more information.

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
