# Peer review of "Three-Dimensional Volume Rendering in Computed Tomography for Evaluation of the Temporomandibular Joint in Dogs"

_animals, 2023, doi:10.3390/ani13203231_

Round 1
Reviewer 1 Report
The images are very beautiful and impressive but the Introduction has many redundant and too much detailed description of CT technique and the Material and Method lack of information about the inclusion and exclusion criteria and the sample (age, breeds, sex, skull conformation, reason for CT study, etc.).
I suggest an extensive English language revision.
Author Response
Dear Editor
Thank you for all your suggestions
We have revised all the manuscript and do all the modifications that concerns the three reviewers. The letter with all modifications is going to be attached in this section.
Our objective has been to perform a descriptive study of the morphological details provided by 3DVR with respect to standard planes. We have tried to choose characteristic images of different cases, both normal and pathologic, to illustrate situations where 3DVR complements and improves the assessment of anatomical details. If the work is accepted this would be Part 1. In a second work, we are finishing a retrospective study reviewing a large number of animals in which we intend to include a statistical study of the different pathologies we have found. In this second work, we will determine and compare the prevalence of the different pathologies assessed independently by two radiologists, with statistical verification of the differences associated to variables such as breed, sex, skull conformation, etc. This first work does not include quantitative variables since its objective is to illustrate and demonstrate with concrete images how 3DVR qualitatively enhances the performance of CT diagnosis.
We have decided to send the work to the English Editing Department of the Journal: [email protected] if the paper would be finally accepted.
Reviewer 1
- The images are very beautiful and impressive but the Introduction has many redundant and too much detailed description of CT technique and the
Thank you for your comment; we have deleted the following paragraphs of the introduction to avoid redundancies:
- Line 64 to 70 of the first manuscript: Axial and helical CT scanners can be differentiated. In the standard axial CT scanners, an X-ray tube was positioned inside the gantry and emitted X-rays when the gantry was rotating around the patient’s body while the table remained stationary [9]. In helical CT scanners, the X-ray tube was continuously rotating using simultaneous and continuous table movement while advancing through the gantry [8]. Helical CT scanners are preferable to axial CT ones.
- Line 73 to 78 of the first manuscript: This system improves CT capability due to its ability to produce a vast quantity of volumetric data in a reduced amount of time, the high resolution, and the ability to create isotropic voxel data and, consequently, reliable multiplanar and three-dimensional (3D) reconstructions [11]. MDCT usually includes helical CT scanners [10]. Nowadays, MDCT scanners have displaced CT ones in human and animal practice although CT initials are commonly used as synonymous.
- Line 79 of the first manuscript: CT (MDCT) images are acquired as transverse slices through the anatomy
- Material and Method lack of information about the inclusion and exclusion criteria and the sample (age, breeds, sex, skull conformation, reason for CT study, etc.).
As we have stated at the beginning of this rebuttal letter, our aim was to make a descriptive study, basically we wanted to illustrate, with specific cases, how 3DVR can improve the morphological details provided by standard CT planes. However we have included the following information about the population used in the study in the Material and Methods section:
- Lines 107-115: This paragraph has been included: We studied a total of 410 dogs: 233 males (56.8%) and 177 females (43.2%) aged between 2 and 12 years. According to head types they were grouped into Mesaticephalic (n=348; 84.87%); Brachycephalic (n=56; 13.65%) and Dolichocephalic (n=6; 1.46%). Dogs underwent CT of head for a variety of reasons, including: nasal cavities and frontal sinuses pathologies (n=88; 21.5%), Orbital and salivary glands (n=13; 3.17%), external, middle and inner ears (n= 38; 93%), calvarium and encephalic (n=187; 45.6%) diseases; oral cavity and masticatory apparatus including temporomandibular joints (n=84; 20.5%). Only 10 animals were studied for specific problems in the temporomandibular joint.
- I suggest an extensive English language revision.
Thank you for your suggestion. We have decided to send the work to the English Editing Department of the Journal: [email protected] if the paper would be finally accepted.

Reviewer 2 Report
My suggestions to the Authors:
- Abstract - the term "Dolicocephalic" change as "Dolichocephalic" - please make this correction within the whole manuscript
- the term: "Fovea pterygodea" change as: "Fovea pterygoidea"
- Figure 2 description - "16) Stylohioid" - change as: "stylohyoid", and please check again the anatomical terms within the whole manuscript
- page 7 line 231 - "stylohioid bone" change as: "stylohyoid bone"
- there is a lack of scale bars within the figures, I suggest to add the scale bars
Author Response
Reviewer 2
- Abstract - the term "Dolicocephalic" change as "Dolichocephalic" - please make this correction within the whole manuscript.
- Done.
- the term: "Fovea pterygodea" change as: "Fovea pterygoidea".
- Done.
- Figure 2 description - "16) Stylohioid" - change as: "stylohyoid", and please check again the anatomical terms within the whole manuscript.
- Done.
- page 7 line 231 - "stylohioid bone" change as: "stylohyoid bone".
- Done.
- There is a lack of scale bars within the figures, I suggest to add the scale bars.
We have used Horos program to image processing. This program for standard-plane CT images shows its proper scales that are very thin. However in 3DVR images this program cannot show any scale bar. In response of your concerns we have tried to draw a 1 cm-scale bar more visible within each image.

Reviewer 3 Report
I thank you for submitting this valuable report. I believe that this report is especially useful because it analyzes CT images of many cases and examines whether detailed evaluation of TMJ can be done using 3DVR images. However, I am particularly concerned about whether objectivity can be assured. The characteristics of individual cases are mentioned and discussed based on unquantified results. The fact that it is unclear how many radiologists who made the diagnosis participated and whether they are qualified also gives the impression that there is little evidence to support the diagnosis. The following is a list of points of particular concern. I hope this will be helpful.
line 125. How many imaging radiologists performed the reading? Did more than one person perform the diagnosis? Can objectivity be guaranteed? Is there no need to verify sensitivity and specificity?
line 130-134. Are there any references for CT imaging conditions?
line 132-133. Was the 0.75 pitch this condition for either 0.6mm or 1.25mm CT slice thickness?
line 135. Was the table not in the ROI? Did you do anything to completely separate the mandible from the table?
line152-153. Please indicate how the lateral view is determined (e.g., with the left and right mandible completely overlapping, etc.)
line161-165. Please explain green line and blue line. Don't these indicate sagittal sections?
line 242-244. Please describe in the results section what differences there are in TMJ characteristics between dolicocephalic and mesaticephalic breeds and brachycephalic breeds.
line 244-246. Figure.7 shows this result?
line342-344. The black arrow is hard to see, please change the color. Also, is one of them rostral and one of them caudal?
line378-381. Please provide results showing this statement.
line398-400. You should provide reasons why it is easier to detect details in a 3DVR image than in a transverse section (e.g., both rostral and caudal sides can be observed in a single image).
line401-402. How many normal dolicocephalic, mesaticephalic, and brachycephalic breeds were in this study? Is it possible to quantify these differences and do a statistical analysis?
line407-408. Is this reason indicated by the following sentence?
line 438-439. Please provide objective evidence for this statement.
line445-446. Is this statement true, there is a technique called 3DMRI?
line 448. I agree with this sentence.
That's all.
Author Response
Reviewer 3
I thank you for submitting this valuable report. I believe that this report is especially useful because it analyzes CT images of many cases and examines whether detailed evaluation of TMJ can be done using 3DVR images. However, I am particularly concerned about whether objectivity can be assured. The characteristics of individual cases are mentioned and discussed based on unquantified results. The fact that it is unclear how many radiologists who made the diagnosis participated and whether they are qualified also gives the impression that there is little evidence to support the diagnosis. The following is a list of points of particular concern. I hope this will be helpful.
- Line 125. How many imaging radiologists performed the reading? Did more than one person perform the diagnosis? Can objectivity be guaranteed? Is there no need to verify sensitivity and specificity?
Thank you for your comment. We have included the following paragraph to address your concerns in lines 144-146: All images were examined independently by two radiologists with more than 15 year experience in performing and reading CT scan. Results were revised for consensus with the participation of a third radiologist.
- Line 130-134. Are there any references for CT imaging conditions?
- Line 120. We have added two studies with TC conditions similar to ours, but not exact, as they deal with other TC models.
- King, A. M.; Diagnostic Imaging of the Tympanic Bulla and Temporomandibular Joint in the Dog, Cat and Rabbit. Doctoral thesis, University of Glasgow, UK, 2008.
- Mielke, B.; Lam R.; Ter Haar, G. Computed Tomographic Morphometry of Tympanic Bulla Shape and Position in Brachycephalic and Mesaticephalic Dog Breeds. Vet Rad Ultrasound 2017; 58, 552-558. doi: 10.1111/vru.12529.
- Line 132-133. Was the 0.75 pitch this condition for either 0.6 mm or 1.25 mm CT slice thickness?
- Lines 1118-120. We have added some information that was omitted in the study. Sequential transverse CT slices of 0.6 mm thickness (spiral pitch factor of 1) and 1.25 mm thickness (spiral pitch factor of 0.75) were obtained.
- Line 135. Was the table not in the ROI? Did you do anything to completely separate the mandible from the table?
- Lines 121-122. This sentence has been included: We placed dogs on sternal decubitus with the mandibles parallel to the table using positional aids.
- Line 152-153. Please indicate how the lateral view is determined (e.g., with the left and right mandible completely overlapping, etc.)
- Lines 138-139. In Figure1 the lateral view of the image was obtained with the left and right mandible completely overlapping.
- Line 161-165. Please explain green line and blue line. Don't these indicate sagittal sections?
-Line 152. This sentence has been included in Figure 1. A right (green line) and left (blue line) sagittal section of the head.
- Line 242-244. Please describe in the results section what differences there are in TMJ characteristics between dolichocephalic and mesaticephalic breeds and brachycephalic breeds.
- Lines 233-234. We have added this sentence to describe the differences in TMJ between dolichocephalic/mesaticephalic dogs and brachycephalic ones: Brachycephalic dogs show flatter TMJ especially in the medial aspect and the retroarticular processes do not encircle the condylar processes.
- Line 244-246. Figure.7 shows this result?
- Lines 242-244. This paragraph has been modified in order to clarify the characteristics of the TMJ: It is also possible to see wide joint spaces (red arrows) and how the condylar processes have scarce osseous densities without a clear definition of its subchondral bones.
- Line 342-344. The black arrow is hard to see, please change the color. Also, is one of them rostral and one of them caudal?
- Lines 338-340: The color of all arrows has been changed to ease the observation. The sentence has also been modified: In C it is easily seen that the most rostral portion of the condylar process (red arrow) is wider than the caudal one (orange arrow).
- Line 378-381. Please provide results showing this statement.
Although the quality of 0.6 and 1.25 mm slices are similar both in human and animal CT scans, in our experience slices thickness of 0.6 mm produce better 3DVR images than 1.25 mm slices and that is the reason why for this paper we have only chosen the 0.6 mm slices in order to have the best images. Thus, the sentence has been removed.
- Line 398-400. You should provide reasons why it is easier to detect details in a 3DVR image than in a transverse section (e.g., both rostral and caudal sides can be observed in a single image).
- Lines 397-398: This sentence has been added: Because you can get the whole vision of the TMJ and both rostral and caudal sides can be observed in a single image.
- Line 401-402. How many normal dolichocephalic, mesaticephalic, and brachycephalic breeds were in this study? Is it possible to quantify these differences and do a statistical analysis?
We have responded to this comment in the Material and Methods section as this question was also raised by other Reviewers. At the beginning of this rebuttal letter we have also tried to explain the primary morphological aim of this first study which we hope will be followed by a second study reporting prevalence of TMJ variations and its association with other variables, such as sex, age, breed, etc.
- Lines 98-103: In response of your comment we have modified the text to show that our objective in this first study was to describe not to compare the morphology of TMJ using both 3DVR images and the 3 standard-plane CT.
- Lines 402-407: this paragraph has been modified to describe the differences of TMJ according to the head type.
- Line 407-408. Is this reason indicated by the following sentence?
In our study, bones of the TMJ in dogs less than 1 year-old were not fully ossified, and an irregular shape could be observed in 3DVR images. These characteristics were detailed in Figure 12, lines 297-300.
- Line 438-439. Please provide objective evidence for this statement.
- Lines 439-441: This sentence has been included: As in the standard planes the morphology of the mandibular fossa may change depending on slice level, 3DVR images could help to address these variations (Figures 2, 4, 5, 11).
- Line 445-446. Is this statement true, there is a technique called 3DMRI?
We have removed the sentence: “but MRI does not have capacity for three-dimensional imaging”.

Round 2
Reviewer 1 Report
The manuscript has significantly improved, however, I think that the Discussion chapter can be better checked for English editing and some minor revisions are needed.
Point by point revision.
Row 31: are they 410 TMJ or 410 dogs? In the Material and Methods chapter you state that the sample was constituted by 410 dogs. If it is right, please, rectify.
Row 62: please, delete “Helical” and replace “CT” with “Computed tomography”.
Rows 70-73: please, replace “Recent investigations have revolutionized the field of CT introducing the multi- 70 detector computed tomography (MDCT) that employs an X-ray tube and a detector array 71 mounted on opposite sides of the patient in a continuously rotating gantry that simultaneously collects imaging data” with “Recently, the introduction of multi-detector computed tomography (MDCT) has deeply improved the volumetric reconstruction due to the real isotropic voxels.” (please, preserve the right references).
Rows 80-81: please, replace “scanning and reconstruction of” with “displaying”.
Row 83: please, replace “projections” with “scans”.
Rows 84-85: please, delete “a data visualization technique that creates”.
Row 88: please, replace “inspection” with “visualization”.
Row 99: after the description of the “manual segmentation technique” maybe it is the case to briefly describe the “automated segmentation technique”.
Row 120: delete “important”.
Row 124: I think that the point 4 is not necessary to this study and therefore it is better to remove it.
Row 124: please, replace “selection” with “provide a pictorial essay”.
Row 125: please, move “some” before “pathological cases” and change “of” with “in”.
Row 131: please, replace “health” with “skull” and replace the term “head” with “skull” throughout the manuscript.
Rows 130-132: please, specify which method was used to classify the skulls and insert a reference study.
Row 148: please, replace “punctured” with “injected”.
Rows 169-171: what kind of evaluations were made by the radiologists?
Row 176: please, replace “of the ramus” with “to the ramus”.
Rows 182-183: delete the phrase.
Row 192: replace “detailed” with “visualization”.
Rows 335-341: in my opinion, soft tissues are better visualized using axial or MPR images and soft tissue window; I think that this point must be removed (included Figure 13).
Row 355: please, rephrase the title in “Pictorial 3DVR essay of some pathological cases”
Rows 487-493: please, delete the period.
Row 506: please, replace “planes of MDCT” with “MPR”.
Author Response
Three Dimensional Volume Rendering in Computed Tomography for Evaluation of the Temporomandibular Joint in Dogs
List of responses to Reviewers’ comments
Thank you very much for your valuable comments in Reports Round 2. We have carefully reviewed all your suggestions regarding the manuscript and we appreciate the opportunity to resubmit the manuscript with revisions.
Letters in blue are the answers of the reviewers’ comments.
REVIEWER 1. REPORT ROUND 2
The manuscript has significantly improved, however, I think that the Discussion chapter can be better checked for English editing and some minor revisions are needed.
Point by point revision.
Row 31: are they 410 TMJ or 410 dogs? In the Material and Methods chapter you state that the sample was constituted by 410 dogs. If it is right, please, rectify.
Thank you for your concerns. We have added this sentence to clarify it: scans of bilateral TMJs of 410 dogs.
Row 62: please, delete “Helical” and replace “CT” with “Computed tomography”.
We have done all your suggestions.
Rows 70-73: please, replace “Recent investigations have revolutionized the field of CT introducing the multi- 70 detector computed tomography (MDCT) that employs an X-ray tube and a detector array 71 mounted on opposite sides of the patient in a continuously rotating gantry that simultaneously collects imaging data” with “Recently, the introduction of multi-detector computed tomography (MDCT) has deeply improved the volumetric reconstruction due to the real isotropic voxels.” (please, preserve the right references).
We have done all your suggestions.
Rows 80-81: please, replace “scanning and reconstruction of” with “displaying”.
We have done all your suggestions.
Row 83: please, replace “projections” with “scans”.
We have done all your suggestions.
Rows 84-85: please, delete “a data visualization technique that creates”.
We have done all your suggestions.
Row 88: please, replace “inspection” with “visualization”.
We have done all your suggestions.
Row 99: after the description of the “manual segmentation technique” maybe it is the case to briefly describe the “automated segmentation technique”.
Thank you for your comment. This sentence has been added: Automated segmentation programs are also available involving placement of a “seed” then expansion of the region to be included or excluded using threshold-based algorithms (12).
Reference 12. Dalrymple, N.C.; Prasad, S.R.; Freckleton, M.W.; Chintapalli, K.N. Introduction to the Language of Three Dimensional Imaging with Multidetector CT. Radiographics 2005, 25, 1409-1428. doi: 10.1148/rg.255055044.
Row 120: delete “important”.
We have done all your suggestions.
Row 124: I think that the point 4 is not necessary to this study and therefore it is better to remove it.
As your concerns, the point 4 has been removed and the objectives have been renumbered.
Row 124: please, replace “selection” with “provide a pictorial essay”.
We have done all your suggestions.
Row 125: please, move “some” before “pathological cases” and change “of” with “in”.
We have done all your suggestions.
Row 131: please, replace “health” with “skull” and replace the term “head” with “skull” throughout the manuscript.
We have done all your suggestions.
Rows 130-132: please, specify which method was used to classify the skulls and insert a reference study.
Thank you for your suggestions. Skull type was classified according to the facial index as previously described (2). This sentence has been modified: According to skull type previously described (2) dogs were grouped into dolichocephalic (n=6, 1.46%); mesaticephalic (n=348, 84.87%);and brachycephalic (n=5; 13.65%).
Reference
- Hermanson, J.W.; De la Hunta, A.; Evans, H.E. Millers and Evans´ Anatomy of the dog, 5th ed.; Elsevier: St. Louis (Missouri), US, 2020; pp. 86-124.
Row 148: please, replace “punctured” with “injected”.
We have done all your suggestions.
Rows 169-171: what kind of evaluations were made by the radiologists?
The evaluation was carried out following stablished criteria for normal (22) and abnormal TMJ (5,6).
References
- King, A. M.; Diagnostic Imaging of the Tympanic Bulla and Temporomandibular Joint in the Dog, Cat and Rabbit. Doctoral thesis, University of Glasgow, UK, 2008.
- Schwarz, T. Temporomandibular Joint and Masticatory Apparatus. In Veterinary Computed Tomography; Schwarz, T., Saunders. , Eds.; Wiley-Blackwell: West Sussex, UK, 2011; pp. 125-136.
- Wisner, E.; Zwingenberger, A. Atlas of small animal CT and MRI. Wiley Blackwell: West Sussex, (UK), 2015. pp. 40-54.
Row 176: please, replace “of the ramus” with “to the ramus”.
We have done all your suggestions.
Rows 182-183: delete the phrase
Unfortunately, we have not found any correspondence with these row numbers in the revised manuscript and thus it was no possible to address these suggestions. Please, would you be so kind to confirm us the exact row numbers? Thank you very much.
Row 192: replace “detailed” with “visualization”.
Unfortunately, we have not found any correspondence with these row numbers in the revised manuscript and thus it was no possible to address these suggestions. Please, would you be so kind to confirm us the exact row numbers? Thank you very much.
Rows 335-341: in my opinion, soft tissues are better visualized using axial or MPR images and soft tissue window; I think that this point must be removed (included Figure 13).
Thank you for your suggestions. The paragraph and the Figure 13 related to the assessment of soft tissues have been removed.
Row 355: please, rephrase the title in “Pictorial 3DVR essay of some pathological cases”
The sentence has been rephrased: 3.4. A pictorial essay of 3DVR images in some pathological cases.
Rows 487-493: please, delete the period.
Unfortunately, we have not found any correspondence with these row numbers in the revised manuscript and thus it was no possible to address these suggestions. Please, would you be so kind to confirm us the exact row numbers? Thank you very much.
Row 506: please, replace “planes of MDCT” with “MPR”.
We have done all your suggestions.

Reviewer 3 Report
Figure 1. Section D and E have green and blue frames, are they really sagittal sections of lines 3 and 4? I just can't understand it.
Author Response
REVIEWER 3. REVIEW REPORT (ROUND 2).
Figure 1. Section D and E have green and blue frames, are they really sagittal sections of lines 3 and 4? I just can't understand it.
Thank you for your concerns. We have modified this paragraph to make it easier to understand: Figure 1. A 6 year-old, female, Golden retriever. Segmentation of a 3DVR image of the head of the dog. Section A: lateral view of the head. Transverse segmentation rostral (red line) and caudal (yellow line) of the ramus of the mandible. Section B: Rostral view of the head obtained at the level of the red line. Section C: Rostral view of the head obtained at the level of the yellow line. Sections D and E represent a magnified image detail of both TMJs obtained at the level of green (3) and blue (4) lines of the section B.
